# Estimation of Health Risks Caused by Metals Contained in E-Cigarette Aerosol through Passive Vaping

**DOI:** 10.3390/toxics11080684

**Published:** 2023-08-09

**Authors:** Wei-Chung Su, Jinho Lee, Kai Zhang, Su-Wei Wong, Anne Buu

**Affiliations:** 1Department of Epidemiology, Human Genetics and Environmental Sciences, School of Public Health, University of Texas Health Science Center at Houston, Houston, TX 77030, USA; 2Department of Environmental Health Sciences, School of Public Health, University at Albany, State University of New York, Rensselaer, NY 12144, USA; 3Department of Health Promotion & Behavioral Sciences, School of Public Health, University of Texas Health Science Center at Houston, Houston, TX 77030, USA

**Keywords:** e-cigarette aerosol, respiratory deposition, metal, passive vaping, health risk

## Abstract

It is expected that secondary exposure to e-cigarette aerosol (passive vaping) will soon become an issue of public health. Passive vaping inhales e-cigarette aerosol containing similar harmful substances as active vaping. However, parallel studies on passive vaping are minimal. Therefore, there is a need for passive vaping-related health risk studies to assess the impact of vaping on public health. This research conducted a series of experiments in a room using a puffing machine and the Mobile Aerosol Lung Deposition Apparatus (MALDA) to study e-cigarette aerosol respiratory deposition through passive vaping. The experimental data acquired were applied to estimate the deposited mass and health risks caused by toxic metals contained in e-cigarette aerosol. Five popular e-cigarette products were used in this study to generate e-cigarette aerosol for deposition experiments. In addition, size-segregated e-cigarette aerosol samples were collected, and metal compositions in the e-cigarette aerosol were analyzed. Results obtained showed that estimated non-cancer risks were all acceptable, with hazard quotient and hazard index all less than 1.0. The calculated cancer risks were also found acceptable, with lifetime excess cancer risk generally less than 1E-6. Therefore, the e-cigarettes tested and the passive vaping exposure scenarios studied do not seem to induce any potential for metal-related respiratory health effects.

## 1. Introduction

E-cigarettes are advertised as harm-reduction products because e-cigarette companies claim that e-cigarettes do not produce carcinogenic substances such as ashes and tar in e-cigarette aerosol [1,2]. Therefore, a social environment favorable to e-cigarettes has gradually been formed with positive attitudes toward friends’ or bystanders’ using e-cigarettes (active vaping) [3]. It was reported that about 70% of young adult e-cigarette users in the U.S. started vaping because e-cigarettes were more acceptable to non-tobacco users than combustible cigarettes [4]. Based on this trend, it is expected that secondary exposure to e-cigarette aerosol (passive vaping) might become an issue with significant impacts on public health [5,6].

However, active vaping is not at all harmless [7]. Studies have found that e-cigarette aerosol contains harmful substances that are associated with many negative physical outcomes, such as changes in heart rate, blood pressure, pulmonary function, etc. [8,9,10]. There are also mental health effects associated with nicotine inhalation including addiction, alertness change, reduced appetite, impulse control, memory, learning and focusing problems, and ADHD symptoms [11,12,13,14,15]. For passive vaping, passive vapers are considered to inhale e-cigarette aerosol containing similar harmful substances as active vapers. However, parallel studies on the health effects of passive vaping are very limited. Given that the published literature on combustible cigarette smoking has consistently found secondhand smokers carry significant health risks [16,17], there is a need for health risk studies focusing on passive vaping to comprehensively assess the impact of vaping on public health.

The substances in e-cigarette aerosol are mainly chemical compounds from e-liquid ingredients and metals from device components in contact with the e-liquid such as the heating coil and battery connectors. Soils contaminated by metals could also result in the presence of metals in the plant and, then, eventually in the e-liquid [18,19]. The size distribution of freshly generated e-cigarette aerosol (active vaping) was reported to range from tens of nanometers to several hundred nanometers in the submicron range [20,21,22]. However, the mode of aged e-cigarette aerosol (passive vaping) was found to be generally around 50 nm in the ultrafine particle size range [23,24,25]. This disparity in e-cigarette aerosol size distribution is mainly due to the natural process of liquid aerosol evaporation [26,27]. E-cigarette aerosols are expected to reduce in size after traveling a distance in the air. Therefore, substances contained in e-cigarette aerosol having low volatility or low vapor pressure, such as metals, will be enriched in the aged e-cigarette aerosol. Once inhaled, these tiny and enriched e-cigarette aerosols can reach and deposit in the lower airways resulting in a considerable deposited mass of toxic substances in the lung. With a routine intake of toxic substances through passive vaping, adverse health effects could be induced.

It is known that certain metals entering the human body through inhalation could induce various non-cancer and cancer adverse effects. For example, the inhalation of Nickel can cause lung inflammation and lung cancer, and exposure to Cadmium can cause decreased lung function and emphysema, as well as lung, trachea, and bronchus cancers [28,29,30]. It has also been reported that positive correlations were found between the metal concentration in the blood and the metal concentration in the aerosol exposed [31]. Since a number of different metals have been found in e-cigarette aerosol [18,32,33], in this study, the respiratory deposited mass of metals through passive vaping was carefully investigated to provide useful information on passive vapers’ health risks. Indeed, with relatively more complete dose–response data on metals in the literature, health risk estimations for metals contained in e-cigarette aerosol are more achievable compared to the estimations for chemical compounds in e-cigarette aerosol. In the future, when more dose–response data on harmful e-cigarette chemicals are available, associated health risk estimations can then be conducted accordingly.

To correctly estimate health risks caused by passive vaping, the key step is to obtain high-quality aerosol respiratory deposition fractions to estimate the associated deposited mass. In the past, it was very challenging to conduct in vitro aerosol respiratory deposition experiments for a complete human respiratory tract. The difficulty was mainly due to the unavailability of human lower airway replicas, such as the lower tracheobronchial (TB) airways and the alveolar region [34,35]. As a result, no representative aerosol respiratory deposition experiments could be conducted, and the nature of aerosol respiratory deposition in human lower airways remains poorly understood. Motivated by the limitations of the conventional aerosol respiratory deposition methods, a Mobile Aerosol Lung Deposition Apparatus (MALDA) was developed. MALDA consists of a set of realistic human airway replicas covering the human upper airways to the lower airways. The MALDA prototype has been applied in several occupational aerosol respiratory deposition experiments, including welding fumes and 3D printing emissions [36,37]. Later, an upgraded MALDA was developed by including the alveolar region and was applied to study e-cigarette aerosol [38,39]. In this study, the upgraded MALDA was further employed to assist in the estimation of health risks caused by passive vaping. A series of aerosol respiratory deposition experiments were conducted in the laboratory. Five e-cigarette products were used in the experiments to generate test e-cigarette aerosol since the characteristics of e-cigarette aerosol such as particle size distributions and metal compositions could be different by the design of the e-cigarette devices and the e-liquid ingredients. These differences can lead to a dissimilar respiratory deposited mass of toxic metals in passive vapers’ airways, causing health risks at different levels.

## 2. Materials and Methods

### 2.1. Mobile Aerosol Lung Deposition Apparatus (MALDA)

MALDA was developed to overcome the critical limitations in conventional lung deposition experimental methods, including tedious research procedures and the exclusion of deeper lung regions. MALDA consists of two major systems: a human airway system and an aerosol measurement system. By placing the two systems on a lab trolley with a battery-powered vacuum pump, MALDA becomes mobile and is capable of carrying out aerosol respiratory deposition experiments in real-life settings under a 30 L/min constant inspiratory flow rate. The human airway system contains a set of 3D-printed realistic human airway replicas covering the human nasal airway, oral airway, throat, trachea, TB airways down to the 11th lung generation, and a representative alveolar region. The human airway system is installed inside a human torso mannequin, as shown in Figure 1a. The aerosol measurement system contains two units of aerosol particle sizer. The particle sizer, Scanning Mobility Particle Sizer (SMPS+C, GRIMM Aerosol Technology, Ainring, Germany), can measure the particle size distribution (number concentration) of aerosol from 7.2 to 272.4 nm with 39 channels. Three sampling probes were designed on the human airway system to allow the particle sizer to measure particle size distributions of inhaled aerosol after they passed major airway regions. With the particle size distribution measured at key airway regions, the aerosol respiratory deposition fraction can be systematically estimated. Figure 1b shows the schematic of MALDA with locations of sampling probes. MALDA and its prototypes have been validated by laboratory aerosol and applied to several environmental and occupational health-related aerosol exposure studies to estimate aerosol respiratory depositions [36,37,38,39,40]. Results obtained from these studies showed that MALDA is a useful tool for efficient on-site aerosol respiratory deposition experiments.

### 2.2. E-Cigarettes and E-Cigarette Aerosol Generation

Based on a survey on e-cigarette usage behaviors among college students (a separate study), four popular e-cigarette devices were selected in this study for passive vaping experiments. The selected e-cigarette products with flavors were Esco Bars (Banana Ice), Kangvape Onee Stick Vintage (Grape Ice), Air Bar Lux (Watermelon Ice), and Puff Bar (Grape). The selection of these e-cigarette products is mainly based on prevalence since they were the most commonly adopted e-cigarette products among the study participants when the selection was made. For the purpose of data comparison, Juul (Menthol) was also adopted in this study because it was once the most popular e-cigarette product in the past.

To generate realistic, repeatable, and representative e-cigarette aerosol for respiratory deposition experiments, an e-cigarette puffing machine (CSM-eSTEP, CH Technologies, Inc., Westwood, NJ, USA) was used. This e-cigarette puffing machine is a user-friendly, portable puffing machine that can be applied to a wide range of e-cigarette products and is suitable for a variety of e-cigarette aerosol exposure studies. The CSM-eSTEP has been used in a vaping-related study to generate representative e-cigarette aerosol [41]. In this study, the e-cigarette puffing machine was operated under a common vaping topography with a puff period of 3 s, puff interval of 30 s, around 200 mL of puff volume (puffing machine output: 4.0 L/min), and a square puff profile shape. The same puffing protocol was applied to all five e-cigarette devices to generate test e-cigarette aerosol.

### 2.3. Passive Vaping Exposure Experiments

The passive vaping experiments were conducted in a room with dimensions of 5.2 m (L) × 3.4 m (W) × 2.4 m (H) and a general ventilation rate of 231 m^3^/h, which is around 5.4 ACH (air change per hour). The e-cigarette puffing machine first generated e-cigarette aerosol at the corner of the room to release e-cigarette aerosol into the room and allow enough time for e-cigarette aerosol freely diffused in the room. Before being released to the room, the generated e-cigarette aerosol passed through a custom-made temperature (T) and relative humidity (RH) conditioner to adjust the status of the aerosol to 37 °C and 98% RH. The use of the T&RH conditioner was to mimic e-cigarette aerosol being exhaled from an active vaper’s airways. After 5 min of e-cigarette aerosol generation, MALDA started measuring aerosol respiratory deposition data. MALDA was placed at a location 6 m away from the e-cigarette puffing machine (diagonal distance), representing a passive vaper exposure to e-cigarette aerosol in an indoor environment. When conducting the experiments, one unit of the particle sizer was always connected to the sampling probe at the inlet of the human airway system (through the nose with the mouth entry closed) to collect the particle size distribution of e-cigarette aerosol that entered the MALDA ($C_{0,d}$). The data collected also represented the size distribution of e-cigarette aerosol in the room. The second unit of the particle sizer took measurements in turns at sampling probes (e.g., TB: $C_{TB,d}$ and alveoli: $C_{Alv,d}$) for obtaining particle size distributions of inhaled e-cigarette aerosol after the aerosol penetrated through major airway regions. Particle size distributions acquired were then used for the calculation of size-dependent aerosol respiratory deposition fractions. In each experiment, three runs of particle size distribution measurements were taken at each sampling probe. At least five experiments were repeated for each e-cigarette product to obtain statistically meaningful averages and standard deviations. Between the two experiments, there was a 30 min waiting interval, and the ventilation system was enhanced to efficiently flush out the e-cigarette aerosol remaining in the room from the previous experiment.

### 2.4. E-Cigarette Aerosol Chemical Composition Analysis

To study metal-induced health risks via passive vaping, information on the toxic metals contained in the e-cigarette aerosol must be available. To obtain size-dependent metal compositions in e-cigarette aerosol, Micro-Orifice Uniform Deposit Impactor (MOUDI, MSP Co., Shoreview, MN, USA) was applied to collect size-segregated e-cigarette aerosol samples. MOUDI is a commercially available multi-state aerosol cascade impactor for collecting aerosol particles from nanometers to micrometers (56 nm to 18 μm) on its 11 impactor stages. Each MOUDI impactor stage has a nominal collectible particle size range under the designed operation flow rate of 30 L/min. Aerosol samples collected on filters can be used for gravimetric analysis and further used for chemical composition analyses. MOUDI has been used recently in active vaping research to study size-dependent substance composition in e-cigarette aerosol [42].

In this study, to collect size-segregated e-cigarette aerosol generated by the five test e-cigarette products using MOUDI, separate chamber exposure studies were carried out. The CSM-eSTEP puffing machine and T&RH conditioner were used again to generate and condition e-cigarette aerosol. The e-cigarette aerosol was delivered into the stainless chamber with a dimension of 1.2 m (L) × 1.2 m (W) × 1.2 m (H) for its main section. A fan was installed in the chamber to enhance the mixing and evaporation of e-cigarette aerosol. After mixing and evaporation in the chamber, the e-cigarette aerosol was further delivered to the MOUDI for size-segregated sample collection. The e-cigarette aerosol generation and sample collection process took a total of 2 h to accumulate an adequate quantity of e-cigarette aerosol samples onto polytetrafluoroethylene (PTFE) membrane filters (PALL Co., Port Washington, NY, USA) placed on the last three MOUDI impactor stages (nominal collectible particle size range: 56–100 nm, 100–180 nm, and 180–320 nm). When the sampling was completed, PTFE filters were unloaded from MOUDI and weighted individually by a microbalance (CAHN-34, ThermoFisher Scientific, Bedford, MA). Filters were then immediately placed in individual glass vials and delivered with blank filters to the ICP Analytical Laboratory and Agilent Facility Center at the University of Houston for chemical analysis. The process of sample collection was replicated three times for every e-cigarette (resulting in a total of nine filters). The collected filters are carefully placed inside 5 mL metal-free conical centrifuge tubes and then filled with ultra-clean double-distilled HNO_3_ and HF. The sealed centrifuge tubes are then placed in an ultrasonic bath for 30 min to assist the acid leach and digestion. After the ultrasonic bath, the centrifuge tubes are placed inside an oven to cook overnight at 80 °C. The digested solutions were carefully verified for complete digestion after cooling, then evaporated to incipient dryness and re-dissolved in 2% HNO_3_ for analysis by inductively coupled plasma mass spectrometry (ICP-MS) technique to acquire the mass of 23 toxic metals contained in the e-cigarette aerosol such as Ni, Cr, Mn, and Cd. The ICP-MS analysis was conducted by Agilent 8800 ICP Triple Quad (ICP-QQQ-MS, Agilent Technologies, Inc., Santa Clara, CA, USA) down to sub-ppb (ng/g) levels with ±5% of precision by following the protocols used in a previous study [43]. The results of the metal composition analysis in e-cigarette aerosol played a key role in estimating the deposited mass, daily dose, and health risks induced by passive vaping.

### 2.5. Deposition Fraction, Deposited Mass, Average Daily Dose, and Health Risk Estimations

After the e-cigarette aerosol size distribution in major human airways was obtained by MALDA, the size-dependent (*d*-dependent) aerosol respiratory deposition fractions in major airway regions were calculated by the following equations:(1)$${DF}_{H+TB,d}\;=\;1-(C_{TB,d}{/C}_{0,d}),$$
(2)$${DF}_{Alv,d}=(C_{TB,d}/C_{0,d})-(C_{Alv,d}/C_{0,d}),$$
(3)$${DF}_{Total,d}=1-(C_{Alv,d}/C_{0,d}),$$
where $C_{0,d}$, $C_{TB,d}$, and $C_{Alv,d}$ are the particle size distributions measured by particle sizers at the inlet, TB airways, and alveolar region, respectively. ${DF}_{H+TB,d}$, ${DF}_{Alv,d}$, and ${DF}_{Total,d}$ are calculated size-dependent aerosol respiratory deposition fraction (values from 0 to 1.0) in Head+TB airways, alveolar region, and the entire human airway system, respectively. By multiplying the size-dependent aerosol respiratory deposition fraction with the size-dependent aerosol mass concentration, the size-dependent deposited mass of e-cigarette aerosol in major human airway regions can then be reasonably calculated:(4)$${DM}_{H+TB,d}={DF}_{H+TB,d}\times M_{d},$$
(5)$${DM}_{Alv,d}={DF}_{Alv,d}\times M_{d},$$
(6)$${DM}_{Total,d}={DF}_{Total,d}\times M_{d},$$
where $M_{d}$ is the e-cigarette aerosol mass concentration (mg/m^3^) by size, which was acquired by the particle size distribution measured by the particle sizer connected at the inlet of the MALDA and applying the density of Glycerol (1.26 g/cm^3^) as the particle density of e-cigarette aerosol. Glycerol (VG) was reported to be the most abundant chemical substance in e-cigarette aerosol [39,42]. ${DM}_{H+TB,d}$, ${DM}_{Alv,d}$, and ${DM}_{Total,d}$ are the size-dependent deposited mass of e-cigarette aerosol (mg/m^3^) in Head+TB airways, the alveolar region, and the entire airway, respectively.

Moreover, from the gravimetric analysis and ICP-MS metal analysis on e-cigarette aerosol collected by MOUDI, size-dependent metal compositions in e-cigarette aerosol were acquired. The deposited mass of metals in the entire human airways can be estimated by:(7)$${DM}_{k}=\Sigma d\;\;{[(DM}_{H+TB,d}+{DM}_{Alv,d})\times \;C_{k,d}],$$
where $C_{k,d}$ is the size-dependent mass ratio (fraction) of a specific metal *k* contained in e-cigarette aerosol. Equation (7) represents the size-cumulative deposited mass of a metal *k* contained in e-cigarette aerosol in passive vapers’ airways. The unit of ${DM}_{k}$ is the mass of metal per cubic meter of air (mg/m^3^). With ${DM}_{k}$ available, the average daily dose (${ADD}_{k}$) and lifetime average daily dose (${LADD}_{k}$) of the harmful metal *k* in passive vapers’ airways can be calculated by:(8)$${ADD}_{k\;}=\;\;{DM}_{k}\times \;Q\times EH\times EF\times {ED}_{A}\;/\;(BW\times \;AT),$$
(9)$${LADD}_{k}=\;{DM}_{k}\times \;Q\times EH\times EF\times {ED}_{L}\;/\;(BW\times \;LT),$$
where Q is the hourly human inhalation rate, which is 0.9 m^3^/h based on human minute ventilation under light activities. EH is the average daily e-cigarette aerosol exposure hours of a passive vaper (h/day), which is assumed to be 4 h/day in this study. EF is the exposure frequency (day/year) indicating the average days in a year that a passive vaper exposes to the e-cigarette aerosol (EF could be reasonably assumed to be 350 day/year). ED is the exposure duration (year) indicating the total years of the passive vaper’s e-cigarette aerosol exposure. For ${ADD}_{k}$ estimation, the exposure duration ${ED}_{A}$ was assumed to be 9 years (central tendency), and for ${LADD}_{k}$ estimation, the exposure duration, ${ED}_{L}$ was assumed to be 30 years (high-end). BW is the average human body weight, which is 70 kg by default. AT in Equation (8) is the averaging time, which could be equal to the exposure duration ${ED}_{A}$. LT in Equation (9) is the life expectancy, which is 70 years by default. With ${ADD}_{k}$ and ${LADD}_{k}$ available, health risks caused by the toxic metal *k* can be assessed. For non-cancer health risks caused by passive vaping, the associated health risks could be estimated by:(10)$${HQ}_{k}={ADD}_{k}/{RfD}_{k},$$
where ${HQ}_{k}$ is the hazard quotient for the harmful metal *k* contained in the e-cigarette aerosol. ${RfD}_{k}$ is the reference dose (RfD) published for the metal *k*. In the case that only the reference concentration (RfC) is available and published, the RfC was converted to RfD using a reasonable inhalation rate of 20 m^3^/day and a default body weight of 70 kg (i.e., RfD = RfC × 20/70). The hazard index (HI) was applied to all toxic metals that can induce similar non-cancer health effects in the lung. The HI can be seen as the summation of all related HQ (i.e., HI =∑ HQ). In the non-cancer risk assessment, when the result of the HQ and HI are found to exceed 1.0, it indicates that the non-cancer adverse health effect is potential for passive vapers via passive vaping.

On the other hand, for cancer risk caused by passive vaping, the lifetime excess cancer risks could be estimated by:(11)$${Cancer\;Risk}_{k}={LADD}_{k}\;\times \;{CSF}_{k},$$
where ${CSF}_{k}$ is the published cancer slope factor (i.e., cancer potency) for the metal *k*. For the cancer risk assessment, when the estimated lifetime excess cancer risk exceeds one in one million (10^−6^), it is considered an unacceptable cancer risk. Published ${RfD}_{k}$, ${RfC}_{k}$, and ${CSF}_{k}$ for selected toxic metals contained in e-cigarette aerosol were collected from EPA (IRIS), ATSDR (MRLs), CalEPA, websites, and related documents [28,29,30].

## 3. Results

### 3.1. MALDA E-Cigarette Aerosol Respiratory Deposition Fractions

Figure 2 expresses the particle size distribution of the e-cigarette aerosol in the room measured by the particle sizer connected at the inlet of the MALDA. E-cigarette aerosol was generated by the puffing machine. Also shown in Figure 2 is the background aerosol measured in the room before generating the e-cigarette aerosol. As can be seen, passive vaping-related e-cigarette aerosol presented a roughly bell-shaped particle size distribution with concentrations several folds higher than the background aerosol. The modes of e-cigarette aerosol were all less than 100 nm within the range of ultrafine particles. E-cigarette aerosol generated by different e-cigarette devices showed different count median diameters (CMD). The measured CMDs from high to low were 81.8 nm for Kangvape (Grape), 75.6 nm for Air Bar Lux (Watermelon), 71.7 nm for Esco Bars (Banana), 59.8 nm for Puff Bar (Grape), and 58.4 nm for JUUL (Menthol). When taking a close look, it can be seen that the particle size distributions of e-cigarette aerosol were centralized within 15 to 200 nm. E-cigarette aerosol outside this range was found to be comparable to the background aerosol. Based on this, the estimation of the respiratory deposition fraction, deposited mass, and health risks in this study were all focused on e-cigarette aerosol within 15 to 200 nm.

Figure 3 shows the comparison of the particle size distribution of e-cigarette aerosol generated by Esco Bars (Banana) w/ and *w*/*o* using the T&RH conditioner. It shows that e-cigarette aerosol passing through a warm and humid environment can increase the size of the e-cigarette aerosol, which could affect the associated deposition fractions in the passive vaper’s airways. Therefore, to generate representative and realistic e-cigarette aerosol for passive vaping experiments, T and RH conditioning should be considered and applied.

Figure 4 shows the size-dependent respiratory deposition fractions of e-cigarette aerosol in the Head+TB airways, alveolar region, and the entire human airways acquired by MALDA. The deposition fractions were calculated based on Equations (1)–(3). Deposition fractions with negative values (unreasonable) were removed from the data set. Dash lines shown in Figure 4 are corresponding conventional lung deposition curves from International Commission on Radiological Protection (ICRP) [44]. It can be seen that respiratory depositions of e-cigarette aerosol basically followed the ICRP curves for all tested e-cigarette devices. There was no noticeable difference in the deposition fraction pattern shown among different e-cigarette products. Although the experimental data showed a slight overestimation in particle size larger than 150 nm, data acquired by MALDA generally agree with the ICRP curves for most of the particle sizes studied. The overestimation might be due to relatively fewer e-cigarette aerosol with particles larger than 150 nm as mentioned above, which might cause uncertainty in associated respiratory deposition fractions.

### 3.2. MOUDI Metal Composition Analysis and Health Risk Estimation

Table 1 shows the mass ratio of metal to e-cigarette aerosol in three MOUDI collectible size ranges. The metal mass is the cumulative mass of 23 metals selected in this study. The mass of the e-cigarette aerosol was acquired by the gravimetric analysis for MOUDI filters after the sample collection, and the mass of metal was obtained by ICP-MS analysis for e-cigarette aerosol on MOUDI filters. It can be seen that the ratio of metal mass to e-cigarette aerosol mass generally increased with the decrease in e-cigarette aerosol size. Figure 5 presents the metal composition of the 23 selected metals in e-cigarette aerosol. The metal composition is presented by relative metal content, which is the ratio of a specific metal mass to the total metal mass measured. As can be seen, Iron (Fe), Chromium (Cr), and Aluminum (Al) are three abundant metals found in e-cigarette aerosol in this study. There was particularly more Nickel (Ni) found in e-cigarette aerosol generated by Esco Bars (Banana) and Puff Bar (Grape) but not in the other three products. Other metals were comparatively less in e-cigarette aerosol, and there seemed no obvious associations and patterns between the metal composition and the aerosol size. Toxic metals found in the e-cigarette aerosol having the potential to cause significant adverse health effects were further selected and used for health risk assessments. Table 2 lists the selected toxic metals of concern and the related health effects that could be induced through the inhalation route. Among these further selected toxic metals, Beryllium (Be), Chromium (Cr VI), Nickel (Ni), Arsenic (As), Cadmium (Cd), and lead (Pb) can cause both non-cancer and cancer effects. The rest of the metals such as Vanadium (V), Manganese (Mn), Cobalt (Co), Molybdenum (Mo), and Antimony (Sb) can mainly cause non-caner effects. With the e-cigarette aerosol respiratory deposition data acquired by MALDA (Figure 4) and with the metal composition data acquired by MOUDI (Figure 5), metal-induced health risks caused by exposure to e-cigarette aerosol through passive vaping can be calculated using Equations (7)–(11). Table 3 lists all non-cancer and cancer health risks estimated based on the passive vaping scenario designed in this study. The complete and detailed data of the calculated deposited mass (DM), average daily dose (ADD), lifetime average daily dose (LADD), reference dose (RfD), cancer slope factor (CSF), and health risks are listed in Table 4. Calculations are based on suitable exposure factors on passive vaping, including an inhalation rate of 0.9 m^3^/h and daily exposure hours of 4 h/day. Values of RfD and CSF used for risk estimation were mainly from the federal agencies, EPA (IRIS) and ATSDR (MRLs). It is worth noting that the mass of Cr obtained from the ICP-MS analysis represents the total chromium. While Cr (VI) is known to be a confirmed human carcinogen, there are no existing studies that have explored the proportion of Cr (VI) within the total chromium content in e-cigarette aerosols. Consequently, it is not possible to determine the correct percentage of Cr (VI) present in the total chromium. Given that Cr (III) and Cr (0) are comparatively less harmful, this study refrained from estimating health risks associated with the Cr found in e-cigarette aerosol. This precaution was taken to avoid any misinterpretation of the results and to prevent drawing conclusions based on incomplete information. Nevertheless, based on the risk estimation shown in Table 3 and Table 4, no potential non-cancer risks (${HQ}_{k}$ < 1.0) were found through the designed passive vaping exposure scenario. After summing up all non-cancer hazard quotients that correspond to the respiratory effects ($\sum {HQ}_{k}$), the calculated hazard index (HI) for all tested e-cigarette products ranged from the lowest 0.0006 for Air Bar Lux (Watermelon) to the highest 0.033 for Esco Bars (Banana), indicating no potential non-cancer risk for adverse respiratory effects (HI < 1.0). On the other hand, the lung cancer risks were also found to be acceptable. Estimated lifetime excess cancer risks were all less than 10^−6^.

## 4. Discussion

As can be seen in Figure 4, the e-cigarette aerosol respiratory deposition was primarily a function of the particle diameter of e-cigarette aerosol and was not significantly affected by the concentration of e-cigarette aerosol. Although the particle size distributions of e-cigarette aerosol were different by e-cigarette products, all deposition data followed a similar pattern close to the ICRP curves. This result might be due to the fact that ICRP curves were established based on compact and spherical particles, and e-cigarette aerosol is also spherical particles (formed by condensation of vaporized e-liquid). Without the particle shape substantially deviating from a sphere, the deposition fraction of e-cigarette aerosol in passive vapers’ airways fairly following the ICRP curve is probable and reasonable. Aerosol with irregular particle shapes, such as agglomerates and fibers, might have respiratory deposition patterns different from the conventional deposition curves to a certain extent. On the other hand, from the viewpoint of estimating the deposited mass of e-cigarette aerosol in airways through passive vaping, it is the concentration of e-cigarette aerosol in the room determines the deposited mass. Given the deposition fraction of a specific aerosol size is roughly constant, a high e-cigarette aerosol concentration (higher particle number) at that particle size will result in more e-cigarette aerosol in that size deposit in passive vapers’ airways. This will increase deposited mass, daily doses, and related health risks as well.

Although all the non-cancer and cancer risks estimated in this study showed acceptable health risks, some exposure factors regarding real-life passive vaping can substantially enhance the concentration of e-cigarette aerosol and then increase the health risks. First, the ventilation rate is an important exposure factor in affecting the e-cigarette concentration and then the health risks. The ventilation rate used in this study (ACH = 5.4) is considered relatively higher than that for general residents (1.0–5.0). With a low ACH in real life, such as in a room with the air conditioner off and windows closed, e-cigarette aerosol would easily accumulate to reach a high concentration. Second, the distance between the active and passive vapers is also an essential exposure factor to decide the concentration of e-cigarette aerosol to which the passive vapers would be exposed. When passive vapers stay very close to active vapers (e.g., much shorter than 6 m in this study), passive vapers will expose and inhale partially diluted e-cigarette aerosol in a high concentration. Third, the concentration of e-cigarette aerosol will be naturally high if there are more vapers using e-cigarettes in the same room, such as in vaping-allowed private lounges or clubs. Therefore, with any one or more of the conditions above occurring in a real-life passive vaping scenario, the concentration of the e-cigarette aerosol would be increased, which will consequently increase related health risks.

In this study, e-cigarette aerosol was measured by two SMPS particle sizers both with a measurement limit of 272.4 nm. Large e-cigarette aerosol such as particles larger than 300 nm may exist in the room to be inhaled by MALDA, but these large particles could not be detected by the particle sizer due to the instrument limit. Theoretically, large aerosol particles would contribute much more deposited mass in human airways. Therefore, it might raise a question in this study regarding overlooking the contribution of larger particles to the deposited mass and the associated health risks because of the instrument limit. However, the focus of deposited mass in this study was on the metals contained in the e-cigarette aerosol, and not actually on the total mass of e-cigarette aerosol. The mass of metals contained in an e-cigarette aerosol particle seemed unchanged while the size of the e-cigarette aerosol shrunk (due to evaporation). The proof of this concept can be seen in Table 1 by the increase in the metal mass ratio in e-cigarette aerosol as the aerosol size decreases. Therefore, larger particles might not possess more metal mass. Moreover, based on the principle of aerosol respiratory deposition, the size-dependent deposition fractions for particles around 300 to 500 nm in the airways are generally lower than 0.1. Thus, based on all the above, with no particular metals contained in the larger particles, and with fewer deposition fractions in the airways for larger particles, the instrument limit is considered not to cause any critical underestimation on the respiratory deposited mass of metals nor on the associated health risks to passive vapers. Nevertheless, there were indeed not many e-cigarette aerosols larger than 200 nm measured in this study, as shown in Figure 2.

The estimation of daily doses and health risks in this study is considered conservative (overestimation) since they were calculated without considering airway clearance and the epithelium absorption rate. It is known that the deposition of e-cigarette aerosol in the inner surface of human lower airways, such as in the TB airways and alveolar region, can all be considered bioavailable for respiratory epithelium absorption to induce potential health effects. Therefore, the results acquired in this study may serve as conservative data with useful information for passive vaping-related health risk assessments. Other specific factors in the experimental design and exposure assumptions can also potentially lead to the overestimations of health risks related to passive vaping. For instance, the puff volume of 200 mL generated by the puffing machine could be relatively higher than what is typically produced by an active vaper using disposable e-cigarettes. Additionally, the assumed 4 h of daily exposure to passive vaping might be considered an overestimation, leading to higher health risks consequently.

Finally, e-cigarette emissions generated from vaping are known to be a mix of aerosol (liquid droplets) and vapor. In this study, the focus was on the metals contained in the aerosol portion. Therefore, health effects and health risks caused by toxic vapors such as formaldehyde, acetaldehyde, and acrolein were not covered. Further research on the vapor portion of e-cigarette emissions is needed to comprehensively understand the health risks caused by passive vaping.

## 5. Conclusions

This research conducted a series of experiments using a puffing machine and MALDA to study e-cigarette aerosol respiratory deposition caused by passive vaping. The obtained experimental data were applied to estimate the deposited mass, daily doses, and health risks caused by toxic metals contained in e-cigarette aerosol. Based on the data acquired, it is the e-cigarette concentration inhaled by the passive vaper that determines the respiratory deposited mass and the related health risks. Therefore, by using a suitable ventilation rate to dilute the e-cigarette aerosol in a passive vaping scenario and by applying enough distance between the active and passive vapers, the e-cigarette aerosol concentration inhaled by the passive vaper could be reasonably decreased. In this way, health risks caused by toxic metals contained in the e-cigarette aerosol could be reduced. Findings from this study may have informative policy implications, such as the e-cigarette product design (reduce metal sources) and recommended distance between active and passive vapers.

## Figures and Tables

**Figure 1 toxics-11-00684-f001:**
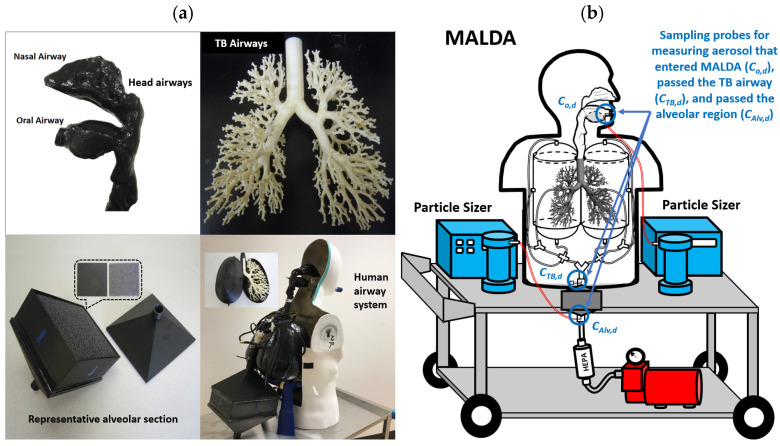
Mobile Aerosol Lung Deposition Apparatus (MALDA): (**a**) the human airway system, and (**b**) the schematic diagram.

**Figure 2 toxics-11-00684-f002:**
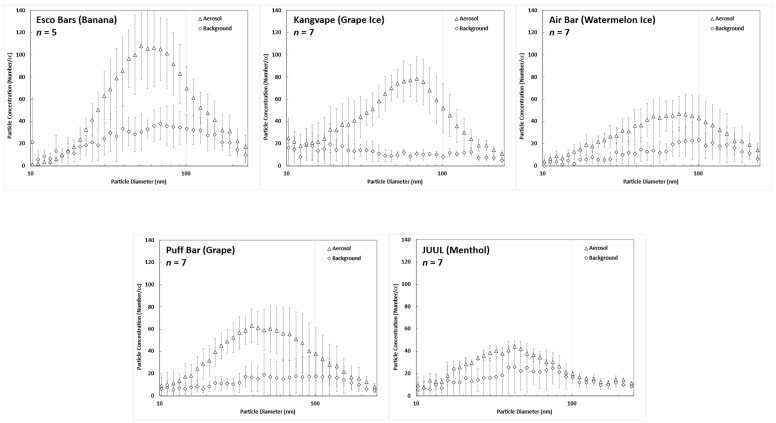
Particle size distributions of e-cigarette aerosol generated by different e-cigarette products, and the background aerosol (error bars in the figures represent the standard deviation of the measurement).

**Figure 3 toxics-11-00684-f003:**
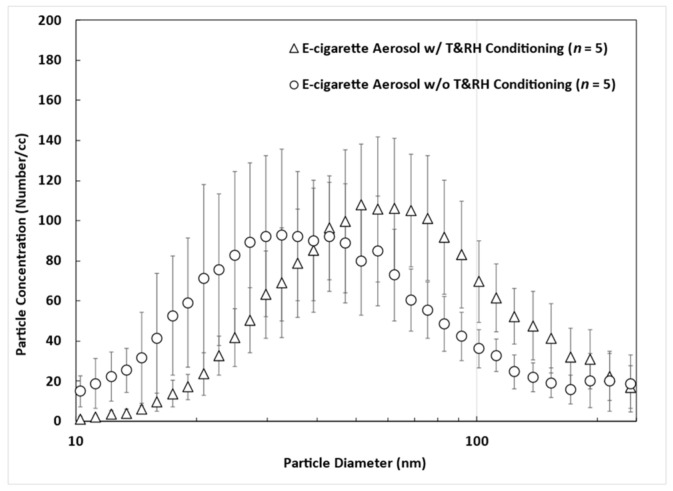
The effect of temperature (T) and relative humidity (RH) conditioning on the particle size distribution of e-cigarette aerosol (error bars represents the standard deviation of the measurement).

**Figure 4 toxics-11-00684-f004:**
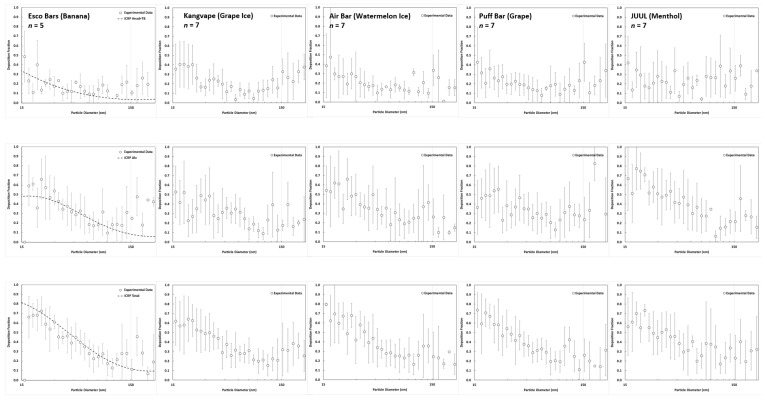
Size-dependent e-cigarette aerosol respiratory deposition fractions in the Head+TB airways, alveolar region, and the entire human airways (from top to down) for five different e-cigarette products (error bars represent the propagated uncertainty of the estimated deposition fractions, and dotted lines represent ICRP conventional curves).

**Figure 5 toxics-11-00684-f005:**
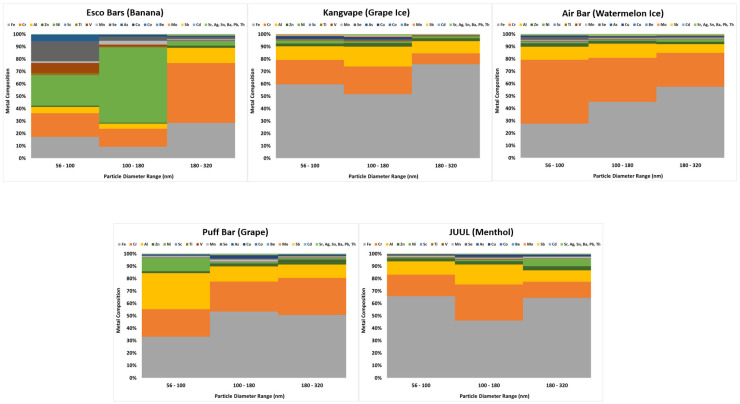
Metal composition in e-cigarette aerosol for metals selected in this study.

**Table 1 toxics-11-00684-t001:** Metal to e-cigarette aerosol mass ratios found by MOUDI analysis (values shown in the parentheses are the total mass of metals measured in mg).

E-Cigarette Product	56–100 nm	100–180 nm	180–320 nm
Esco Bars (Banana)	0.061 ± 0.028 (3 × 10^−4^ ± 2 × 10^−4^)	0.023 ± 0.005 (2 × 10^−4^ ± 7 × 10^−5^)	0.018 ± 0.001 (2 × 10^−4^ ± 3 × 10^−5^)
Kangvape Vintage (Grape)	0.038 ± 0.005 (5 × 10^−4^ ± 4 × 10^−7^)	0.033 ± 0.001 (4 × 10^−4^ ± 3 × 10^−5^)	0.025 ± 0.016 (6 × 10^−4^ ± 3 × 10^−4^)
Air Bar Lux (Watermelon)	0.033 ± 0.009 (2 × 10^−4^ ± 3 × 10^−5^)	0.027 ± 0.005 (3 × 10^−4^ ± 9 × 10^−6^)	0.016 ± 0.001 (2 × 10^−4^ ± 5 × 10^−5^)
Puff Bar (Grape)	0.119 ± 0.146 (4 × 10^−4^ ± 4 × 10^−4^)	0.055 ± 0.032 (5 × 10^−4^ ± 4 × 10^−4^)	0.039 ± 0.001 (6 × 10^−4^ ± 9 × 10^−5^)
JUUL (Menthol)	0.067 ± 0.043 (7 × 10^−4^ ± 2 × 10^−4^)	0.123 ± 0.055 (6 × 10^−4^ ± 4 × 10^−4^)	0.088 ± 0.014 (5 × 10^−4^ ± 2 × 10^−4^)

**Table 2 toxics-11-00684-t002:** Selected toxic metals and associated health effects through inhalation exposure.

Metal	Non-Cancer Effects *	Cancer Effects ^!^ (Carcinogen Classification)
Beryllium (Be)	Beryllium sensitization and chronic beryllium disease	Lung Cancer (EPA:B1; IARC:1)
Vanadium (V)	Lung damage	–
Chromium (Cr VI)	Lactate dehydrogenase in bronchioalveolar lavage fluid	Lung Cancer (EPA:A; IARC:1)
Manganese (Mn)	Impaired lung and neurobehavioral functions	–
Cobalt (Co)	Asthma-like allergy and decreased lung function	(IARC:2B)
Nickel (Ni)	Lung inflammation	Lung Cancer (EPA:A; IARC:1)
Arsenic (As)	Respiratory irritation	Lung Cancer (EPA:A; IARC:1)
Molybdenum (Mo)	Nasal lesions	–
Cadmium (Cd)	Emphysema and decreased lung function	Lung Cancer (EPA:B1; IARC:1)
Antimony (Sb)	Pneumoconiosis and laryngitis	–
Lead (Pb)	Altered neurosensory function	(EPA:B2; IARC:2B)

* U.S. Agency for Toxic Substances and Disease Registry (ATSDR) ToxGuide^TM !^ US EPA Integrated Risk Information System (IRIS).

**Table 3 toxics-11-00684-t003:** Health risks caused by metals contained in e-cigarette aerosol through passive vaping: (**a**) non-cancer, and (**b**) cancer.

**(a)**									
**E-Cigarette Product**	**Be**	**V**	**Mn**	**Co**	**Ni**	**Mo**	**Cd**	**Sb**	**HI ^#^**
Esco Bars (Banana)	9.1 × 10^−5^	1.3 × 10^−3^	2.8 × 10^−3^	6.6 × 10^−5^	2.9 × 10^−2^	1.3 × 10^−6^	2.1 × 10^−5^	2.9 × 10^−6^	**3.3 × 10^−2^**
Kangvape Vintage (Grape)	5.2 × 10^−5^	3.8 × 10^−5^	4.5 × 10^−4^	2.1 × 10^−5^	5.5 × 10^−4^	1.4 × 10^−6^	1.3 × 10^−4^	5.7 × 10^−6^	**1.3 × 10^−3^**
Air Bar Lux (Watermelon)	3.6 × 10^−5^	2.5 × 10^−5^	2.5 × 10^−4^	8.8 × 10^−6^	2.2 × 10^−4^	5.9 × 10^−7^	4.3 × 10^−6^	1.4 × 10^−5^	**5.6 × 10^−4^**
Puff Bar (Grape)	4.7 × 10^−5^	3.6 × 10^−5^	7.6 × 10^−4^	4.8 × 10^−5^	1.2 × 10^−3^	1.3 × 10^−6^	1.4 × 10^−5^	2.3 × 10^−6^	**2.1 × 10^−3^**
JUUL (Menthol)	1.0 × 10^−4^	6.2 × 10^−5^	2.5 × 10^−3^	1.0 × 10^−4^	5.5 × 10^−3^	3.5 × 10^−6^	6.8 × 10^−5^	3.9 × 10^−6^	**8.3 × 10^−3^**
**(b)**									
**E-Cigarette Product**	**Be**	**Ni**	**As**	**Cd**	**Pb**				
Esco Bars (Banana)	1.9 × 10^−9^	2.7 × 10^−7^	1.3 × 10^−7^	1.6 × 10^−10^	1.2 × 10^−11^				
Kangvape Vintage (Grape)	1.1 × 10^−9^	5.1 × 10^−9^	3.3 × 10^−9^	1.0 × 10^−9^	1.5 × 10^−11^				
Air Bar Lux (Watermelon)	7.4 × 10^−10^	2.0 × 10^−9^	2.2 × 10^−9^	3.3 × 10^−11^	7.0 × 10^−12^				
Puff Bar (Grape)	9.7 × 10^−10^	1.1 × 10^−8^	3.7 × 10^−9^	1.1 × 10^−10^	2.9 × 10^−11^				
JUUL (Menthol)	2.1 × 10^−9^	5.1 × 10^−8^	6.2 × 10^−9^	5.3 × 10^−10^	4.5 × 10^−11^				

^#^ HI was calculated by adding up non-cancer risks on the respiratory system caused by Be, V, Mn, Co, Ni, Cd, and Sb.

**Table 4 toxics-11-00684-t004:** Calculated daily deposited mass, average daily dose, lifetime average daily dose, reference dose, cancer slope factor, non-cancer risk, and cancer risk.

Metal	E-Cigarette Product	Daily Deposited mass^!^ (mg/day)	Non-Cancer			Cancer		
			ADD ^#^ (mg/kg−day)	RfD * (mg/kg−day)	Non-Cancer Risk (HQ)	LADD ^$^ (mg/kg−day)	CSF * (mg/kg−day)^−1^	Cancer Risk
**Be**	Esco Bars (Banana)	3.8 × 10^−8^	5.2 × 10^−10^	5.7 × 10^−6^	9.1 × 10^−5^	2.2 × 10^−10^	8.4	1.9 × 10^−9^
	Kangvape Vintage (Grape)	2.2 × 10^−8^	3.0 × 10^−10^	5.7 × 10^−6^	5.2 × 10^−5^	1.3 × 10^−10^	8.4	1.1 × 10^−9^
	Air Bar Lux (Watermelon)	1.5 × 10^−8^	2.0 × 10^−10^	5.7 × 10^−6^	3.6 × 10^−5^	8.8 × 10^−11^	8.4	7.4 × 10^−10^
	Puff Bar (Grape)	2.0 × 10^−8^	2.7 × 10^−10^	5.7 × 10^−6^	4.7 × 10^−5^	1.1 × 10^−10^	8.4	9.7 × 10^−10^
	JUUL (Menthol)	5.9 × 10^−10^	5.9 × 10^−10^	5.7 × 10^−6^	1.0 × 10^−4^	2.5 × 10^−10^	8.4	2.1 × 10^−9^
**V**	Esco Bars (Banana)	2.7 × 10^−6^	3.7 × 10^−8^	2.9 × 10^−5^	1.3 × 10^−3^	1.6 × 10^−8^	−	−
	Kangvape Vintage (Grape)	7.9 × 10^−8^	1.1 × 10^−9^	2.9 × 10^−5^	3.8 × 10^−5^	4.7 × 10^−10^	−	−
	Air Bar Lux (Watermelon)	5.1 × 10^−8^	7.0 × 10^−10^	2.9 × 10^−5^	2.5 × 10^−5^	3.0 × 10^−10^	−	−
	Puff Bar (Grape)	7.5 × 10^−8^	1.0 × 10^−9^	2.9 × 10^−5^	3.6 × 10^−5^	4.4 × 10^−10^	−	−
	JUUL (Menthol)	1.3 × 10^−7^	1.8 × 10^−9^	2.9 × 10^−5^	6.2 × 10^−5^	7.6 × 10^−10^	−	−
**Mn**	Esco Bars (Banana)	2.9 × 10^−6^	3.9 × 10^−8^	1.4 × 10^−5^	2.8 × 10^−3^	1.7 × 10^−8^	−	−
	Kangvape Vintage (Grape)	4.7 × 10^−7^	6.5 × 10^−9^	1.4 × 10^−5^	4.5 × 10^−4^	2.8 × 10^−9^	−	−
	Air Bar Lux (Watermelon)	2.6 × 10^−7^	3.6 × 10^−9^	1.4 × 10^−5^	2.5 × 10^−4^	1.5 × 10^−9^	−	−
	Puff Bar (Grape)	7.9 × 10^−7^	1.1 × 10^−8^	1.4 × 10^−5^	7.6 × 10^−4^	4.7 × 10^−9^	−	−
	JUUL (Menthol)	2.6 × 10^−6^	3.5 × 10^−8^	1.4 × 10^−5^	2.5 × 10^−3^	1.5 × 10^−8^	−	−
**Co**	Esco Bars (Banana)	1.4 × 10^−7^	1.9 × 10^−9^	2.9 × 10^−5^	6.6 × 10^−5^	8.1 × 10^−10^	−	−
	Kangvape Vintage (Grape)	4.3 × 10^−8^	5.9 × 10^−10^	2.9 × 10^−5^	2.1 × 10^−5^	2.5 × 10^−10^	−	−
	Air Bar Lux (Watermelon)	1.8 × 10^−8^	2.5 × 10^−10^	2.9 × 10^−5^	8.8 × 10^−6^	1.1 × 10^−10^	−	−
	Puff Bar (Grape)	1.0 × 10^−7^	1.4 × 10^−9^	2.9 × 10^−5^	4.8 × 10^−5^	5.9 × 10^−10^	−	−
	JUUL (Menthol)	2.2 × 10^−7^	3.0 × 10^−9^	2.9 × 10^−5^	1.0 × 10^−4^	1.3 × 10^−9^	−	−
**Ni**	Esco Bars (Banana)	5.5 × 10^−5^	7.5 × 10^−7^	2.6 × 10^−5^	2.9 × 10^−2^	3.2 × 10^−7^	0.84	2.7 × 10^−7^
	Kangvape Vintage (Grape)	1.0 × 10^−6^	1.4 × 10^−8^	2.6 × 10^−5^	5.5 × 10^−4^	6.1 × 10^−9^	0.84	5.1 × 10^−9^
	Air Bar Lux (Watermelon)	4.1 × 10^−7^	5.6 × 10^−9^	2.6 × 10^−5^	2.2 × 10^−4^	2.4 × 10^−9^	0.84	2.0 × 10^−9^
	Puff Bar (Grape)	2.3 × 10^−6^	3.2 × 10^−8^	2.6 × 10^−5^	1.2 × 10^−3^	1.4 × 10^−8^	0.84	1.1 × 10^−8^
	JUUL (Menthol)	1.0 × 10^−5^	1.4 × 10^−7^	2.6 × 10^−5^	5.5 × 10^−3^	6.1 × 10^−8^	0.84	5.1 × 10^−8^
**As**	Esco Bars (Banana)	1.5 × 10^−6^	2.1 × 10^−8^	−	−	9.0 × 10^−9^	15.05	1.3 × 10^−7^
	Kangvape Vintage (Grape)	3.7 × 10^−8^	5.1 × 10^−10^	−	−	2.2 × 10^−10^	15.05	3.3 × 10^−9^
	Air Bar Lux (Watermelon)	2.4 × 10^−8^	3.3 × 10^−10^	−	−	1.4 × 10^−10^	15.05	2.2 × 10^−9^
	Puff Bar (Grape)	4.2 × 10^−8^	5.8 × 10^−10^	−	−	2.5 × 10^−10^	15.05	3.7 × 10^−9^
	JUUL (Menthol)	7.1 × 10^−8^	9.7 × 10^−10^	−	−	4.1 × 10^−10^	15.05	6.2 × 10^−9^
**Mo**	Esco Bars (Banana)	5.6 × 10^−8^	7.7 × 10^−10^	5.7 × 10^−4^	1.3 × 10^−6^	3.3 × 10^−10^	−	−
	Kangvape Vintage (Grape)	6.0 × 10^−8^	8.2 × 10^−10^	5.7 × 10^−4^	1.4 × 10^−6^	3.5 × 10^−10^	−	−
	Air Bar Lux (Watermelon)	2.5 × 10^−8^	3.4 × 10^−10^	5.7 × 10^−4^	5.9 × 10^−7^	1.4 × 10^−10^	−	−
	Puff Bar (Grape)	5.5 × 10^−8^	7.6 × 10^−10^	5.7 × 10^−4^	1.3 × 10^−6^	3.2 × 10^−10^	−	−
	JUUL (Menthol)	1.5 × 10^−7^	2.0 × 10^−9^	5.7 × 10^−4^	3.5 × 10^−6^	8.6 × 10^−10^	−	−
**Cd**	Esco Bars (Banana)	4.3 × 10^−9^	5.9 × 10^−11^	2.9 × 10^−6^	2.1 × 10^−5^	2.5 × 10^−11^	6.3	1.6 × 10^−10^
	Kangvape Vintage (Grape)	2.7 × 10^−8^	3.7 × 10^−10^	2.9 × 10^−6^	1.3 × 10^−4^	1.6 × 10^−10^	6.3	1.0 × 10^−9^
	Air Bar Lux (Watermelon)	9.0 × 10^−10^	1.2 × 10^−11^	2.9 × 10^−6^	4.3 × 10^−6^	5.3 × 10^−12^	6.3	3.3 × 10^−11^
	Puff Bar (Grape)	3.0 × 10^−9^	4.1 × 10^−11^	2.9 × 10^−6^	1.4 × 10^−5^	1.7 × 10^−11^	6.3	1.1 × 10^−10^
	JUUL (Menthol)	1.4 × 10^−8^	1.9 × 10^−10^	2.9 × 10^−6^	6.8 × 10^−5^	8.3 × 10^−11^	6.3	5.3 × 10^−10^
**Sb**	Esco Bars (Banana)	1.8 × 10^−8^	2.4 × 10^−10^	8.6 × 10^−5^	2.9 × 10^−6^	1.0 × 10^−10^	−	−
	Kangvape Vintage (Grape)	3.6 × 10^−8^	4.9 × 10^−10^	8.6 × 10^−5^	5.7 × 10^−6^	2.1 × 10^−10^	−	−
	Air Bar Lux (Watermelon)	8.6 × 10^−8^	1.2 × 10^−9^	8.6 × 10^−5^	1.4 × 10^−5^	5.1 × 10^−10^	−	−
	Puff Bar (Grape)	1.4 × 10^−8^	2.0 × 10^−10^	8.6 × 10^−5^	2.3 × 10^−6^	8.5 × 10^−11^	−	−
	JUUL (Menthol)	2.4 × 10^−8^	3.3 × 10^−10^	8.6 × 10^−5^	3.9 × 10^−6^	1.4 × 10^−10^	−	−
**Pb**	Esco Bars (Banana)	4.8 × 10^−8^	6.6 × 10^−10^	−	−	2.8 × 10^−10^	0.042	1.2 × 10^−11^
	Kangvape Vintage (Grape)	6.0 × 10^−8^	8.2 × 10^−10^	−	−	3.5 × 10^−10^	0.042	1.5 × 10^−11^
	Air Bar Lux (Watermelon)	2.8 × 10^−8^	3.9 × 10^−10^	−	−	1.7 × 10^−10^	0.042	7.0 × 10^−12^
	Puff Bar (Grape)	1.2 × 10^−7^	1.6 × 10^−9^	−	−	6.8 × 10^−10^	0.042	2.9 × 10^−11^
	JUUL (Menthol)	1.8 × 10^−7^	2.5 × 10^−9^	−	−	1.1 × 10^−9^	0.042	4.5 × 10^−11^

^!^ Daily deposited mass (DM_k_ × Q × EH) was calculated based on the cumulative respiratory deposited mass of a metal (mg/m^3^) multiplied by 0.9 m^3^/h inhalation rate and 4 h/daily passive vaping hours as stated in Equations (8) and (9). * RfD and CSF used for risk estimation were mainly obtained from EPA (IRIS) and ATSDR (MRLs). ^#^ ADD was calculated based on 9 years of exposure. ^$^ LADD was calculated based on 30 years of exposure.

## Data Availability

Some or all data that support the findings of this study are available from the corresponding author upon reasonable request.

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
