# Peer review of "Estimation of Health Risks Caused by Metals Contained in E-Cigarette Aerosol through Passive Vaping"

_toxics, 2023, doi:10.3390/toxics11080684_

Round 1

Reviewer 1 Report

Dear Authors,

It is my pleasure to go through the research and suggest modifications aimed at overall improvement of the article for readers. 

1. Introduction must have a section on impact of metal pollution/exposure on human health. And most of the heavy metals need to be mentioned. Pollution may happen due to transportation (https://www.sciencedirect.com/science/article/abs/pii/S0141113623001708), food crops (https://www.sciencedirect.com/science/article/pii/S0160412018327971), burning of waste (https://www.sciencedirect.com/science/article/pii/S0160412018312662), energy sector (https://www.sciencedirect.com/science/article/abs/pii/S2214785323034119)etc. After mentioning the possible pollution and exposure present for heavy metal, they can add that 'vaping' is a less investigated exposure pathway and hence they are focusing on that. This will invariably introduce the novelity of this article to readers after building a context.

2. Authors should mention some statistical use in the method section and what is the 'n' value? How Many times the experiments were conducted? 

3. Metal particle distribution has been mentioned but as concentration of metal is required for Health risk analysis model, that also need to be reported with statistical analysis such as in articles, https://www.tandfonline.com/doi/abs/10.1080/10807039.2017.1415131, https://www.tandfonline.com/doi/full/10.1080/23311843.2016.1193110 etc.

Overall, I find this article novel but these insights can improve the manuscript for the better.

English language is fine but can be improved 

Author Response

Please see the point-to-point response in the attachment. Thanks!

Reviewer 2 Report

I agree that data on passive e-cigarette aerosol is lacking. I do think that without sidestream aerosol (sidestream would be present in cigarette smoke) that the exposure and risk would be low. Also, although the reasoning for using the e-cigarettes from this study is good, they are not e-cigarettes that are used for producing large vapor clouds that may be of greater risk. A good point is made that people may be more likely to experience passive aerosol exposure because e-cigarettes may be perceived as less harmful than cigarette smoke. Overall, I like that the MALDA is a unique collection device that was made to solve the problem of no proper available model to mimic lower airways.

I have concerns and comments below about the metal data. If this is a publication that focuses on health effects from metals, the method for preparing and analyzing the samples for metals should be presented. This should include a filter blank. If chromium speciation was also performed, that needs to be stated and presented. If not, it cannot be assumed that the total chromium is chromium (VI) to make your conclusions. It is very likely not. It is a bold and inaccurate statement to claim cancer risk from chromium (VI) that is most likely not even present and provide no evidence for chromium (VI) as opposed to chromium (III). I also do not see anywhere to explain that you chose to assume that all of the chromium detected was Cr (VI) with no Cr (III) which would be an overestimation. You lead the reader to believe that the aerosol contains Cr (VI) at harmful levels.

Abstract

Remove the conclusion about chromium (VI): “cancer risks caused by chromium (Cr VI) contained in e-cigarette aerosol.” There is no evidence that chromium (VI) was found in the aerosol. Total chromium does not assume to be 100% chromium (VI).

Introduction:

Line 51: Metals are coming from device components in contact with the e-liquid, not just the heating coil. Change this to say device components and then could list heating coil, battery connectors, solder, etc.

Materials & Methods:

Line 143: How was it decided to use a 200 mL puff volume? The ISO/Coresta standard puff regimen for e-cigarettes is 55 mL. Although the 55 mL is probably a lower volume than real users vape, 200 mL seems very high. It is especially high because I believe all of these devices contain nicotine salts and are not intended for large volume vapor cloud production.

Interesting use of temperature and humidity to mimic exhaled breath.

I agree with the choice to have the particle sizer at the beginning of the airway (how most other studies would probably look at passive aerosol particles) and then within the MALDA.

Agree with 5 replicates of the experiment for each e-cigarette.

Sample collection for MOUDI for metals was 2 hours-2 hours of aerosol generation/puffing or just 2 hours of air collection after the 5 minutes of the puffing regimen?

MOUDI cascade impactor with PTFE filters: Is there a blank for the filters that is subtracted for metals? Preferably a blank that is also put in the MOUDI with room air and then analyzed the same way the sample filters are and subtracted.

3 filter sets per cigarette means 3 replicates collecting 3 filters each time?

Placed in glass vials-how were they then prepared? There needs to be information on sample preparation of the filters for analysis. Were they soaked, vortexed, microwave digested? What is the acid solution?

ICP-MS analysis: What is the instrument used? What cell gases if any? Calibration curve? LOD, LRL? Semi-quantitative or quantitative? Sample prep and instrumental analysis information needs to be presented in paper or supplemental material.

From calculations, it seems like you are assuming a passive user to be someone who lives with someone who vapes: 4 hours/day for 350 days of exposure is quite often.

As as a note, the RfDs should be based on inhalation (not ingestion, etc.) and not on chronic/constant inhalation.

Results

Line 336: Typo

“only the calculated cancer risk caused by Chromium (Cr (VI)) was shown to be unacceptable, with the estimated risk slightly higher than 10−6.”: Again, this is not an appropriate conclusion. You have total chromium data, not chromium (VI) data. All of the total chromium is most likely Cr (III). If you want to make this statement, you need to have chromium speciation data.

Discussion

Discussion mentions how the exposure could actually be higher and lists reasons but also could list more reasons it could be lower. There is the limitation stated about the airway clearance/absorption. But could also mention: The 200 mL puff volume could be an overestimation of puff volume/especially with these disposable type devices. Could the temperature and humidity adjuster alter the particles/aerosol in a way that is not actually expected from an exhaled breath? The 4 hours/day of passive vaping exposure can also be an overestimation. If RfDs were based off of chronic exposure, etc., that should also be mentioned as a limitation.

Author Response

(The authors gave the same response as above.)

Round 2

Reviewer 1 Report

This article can be accepted in the present form.

English language is fine.

Author Response

The author appreciates the reviewer's comments.

Reviewer 2 Report

Thank you for thoroughly addressing all of my comments and concerns. In particular, I appreciate the rewording of chromium statements and additional method details for metals analysis. I may have missed it but if blank filters were used for the metal analysis, that should be stated. If not, I highly recommend preparing and analyzing blank filters for background subtraction for any further metal analysis you plan on performing. I noticed one new typo:

Line 401: Typo “Table 2 Table 1”

Author Response

The author appreciates the reviewer's comments and the manuscript has been revised accordingly.

1. The blank filters are now mentioned on Page 5 Line 201 with gray highlights.  

2. Texts that cite Table 1 and Table 2 have been carefully checked.

Table 1 is for the metal mass ratio, which is on Page 10.

Table 2 is for the metal toxicity (health effects), which is on Page 11.